# On the Mysterious Optimization Geometry of Deep Neural Networks

## Abstract

Understanding why gradient-based algorithms are successful in practical deep learning optimization is a fundamental and long-standing problem. Most existing works promote the explanation that deep neural networks have smooth and amenable nonconvex optimization geometries. In this work, we argue that this may be an oversimplification of practical deep learning optimization by revealing a mysterious and complex optimization geometry of deep networks through extensive experiments. Specifically, we consistently observe two distinct geometric patterns in training various deep networks: a regular smooth geometry and a mysterious zigzag geometry, where gradients computed in adjacent iterations are extremely negatively correlated. Also, such a zigzag geometry exhibits a fractal structure in that it appears over a wide range of geometrical scales, implying that deep networks can be highly non-smooth in certain local parameter regions. Moreover, our results show that a substantial part of the training progress is achieved under such complex geometry. Therefore, the existing smoothness-based explanations do not fully match the practice.

## 1 Introduction

Training simple neural networks is known to be an NP-complete problem (Blum & Rivest, 1988). However, in modern machine learning, training deep neural networks turns out to be incredibly easy in that many simple gradient-based optimization algorithms can consistently achieve low loss (Robbins, 2007; Kingma & Ba, 2015; Duchi et al., 2011). Such an observation inspires researchers to think that there might be specific simple structures of deep neural networks that make nonconvex optimization easy and tractable. Many works have been developed in the past decade to seek justifiable explanations, either theoretically or empirically.

Specifically, from a theoretical perspective, many nonconvex optimization theories have been developed to explain the success of deep learning optimization. The key idea is to prove that deep neural networks have certain nice geometries that guarantee convergence to the global minimum in nonconvex optimization. For example, many types of deep neural networks such as over-parameterized residual networks (Zhang et al., 2019; Allen-Zhu et al., 2019b; Du et al., 2019a), recurrent networks (Allen-Zhu et al., 2019a), nonlinear networks (Zou et al., 2020; Zhou et al., 2016), and linear networks (Frei & Gu, 2021; Zhou & Liang, 2017) have been shown to satisfy the so-called gradient dominant geometry (Karimi et al., 2016). On the other hand, shallow ReLU networks (Soltanolkotabi, 2017; Zhong et al., 2017; Fu et al., 2019; Du et al., 2019b), deep residual networks (Du et al., 2019a), and some nonlinear networks (Mei et al., 2018; Du & Lee, 2018) have been shown to satisfy the local strong convexity geometry. Both geometry types guarantee the convergence of gradient-based algorithms to a global minimum at a linear rate.

From an empirical perspective, researchers have found that skip connections and batch normalization of deep networks can substantially improve the smoothness of the optimization geometry (Li et al., 2018a; Santurkar et al., 2018; Zhou et al., 2019). Furthermore, some other works found that there is a continuous low-loss path between the minima of deep networks (Verpoort et al., 2020; Draxler et al., 2018). In particular, it is observed that a simple linear interpolation between the initialization point and global optimum encounters no significant barrier for many deep networks (Goodfellow

et al., 2015). Moreover, many networks have been shown to possess wide and flat minima that tend to generalize well (Hao et al., 2019; Mulayoff & Michaeli, 2020).

Despite the comprehensiveness of these existing works, they all aim to promote nice geometries to demystify the success of deep learning optimization. While this is an important step toward understanding deep learning, sometimes the results and conclusions can be illusional: the theories and empirical evidence may not necessarily reflect the underlying challenge of optimization in practical deep learning. The success of deep learning optimization may be due to complicated unknown mechanisms oversimplified by existing works. This constitutes **the goal of this paper** – to investigate the optimization geometry of deep networks and reveal its complex and mysterious geometric patterns that may challenge the existing perception of deep learning optimization.

## 1.1 OUR CONTRIBUTIONS

We apply the full batch gradient descent to train various popular deep networks on different datasets and study the geometry along the optimization trajectory via gradient correlation-based metrics (defined in Section 2). Specifically, we observe the following distinct geometric patterns.

- We consistently observe two distinct geometric patterns in all the experiments: (i) smooth geometry where gradients computed in adjacent iterations are highly positively correlated (in terms of the cosine similarity defined in eq. (2)) and point toward similar directions, and (ii) *mysterious zigzag geometry* where gradients computed in adjacent iterations are highly negatively correlated and point toward opposite directions. Interestingly, we find that for convolutional networks, the training starts with the smooth geometry and transfers to the zigzag geometry later on. On the contrary, the training of residual networks starts with the zigzag geometry and transfers to the smooth geometry afterward. Moreover, a substantial part of the training loss decrease is attained under the complex zigzag geometry in all of the experiments.

- We further investigate the mysterious zigzag geometry of deep networks and find that it has a complex *fractal structure*. Specifically, when we zoom into the local geometry by training the deep networks with very small learning rates, we still observe the same zigzag geometry. It shows that the local geometry of deep networks can be highly non-smooth in a wide range of geometrical scales. Moreover, the zigzag geometric pattern tends to be stronger when we zoom into a smaller geometrical scale. These observations challenge the existing explanations of deep learning optimization based on smooth-type geometries.

- Based on the local statistics of mean gradient correlation, we propose a low-cost *geometry-adapted* warm-up learning rate scheduling scheme for large-batch training of residual networks. We show that it leads to comparable convergence speed and test performance to those of the original heuristic version with parameter fine-tuning.

## 2 PRELIMINARIES ON GRADIENT CORRELATION

To understand the optimization geometry in deep learning, we propose investigating the gradients along the optimization trajectory generated by full-batch gradient descent. We consider full-batch gradient descent as it is noiseless and reflects the exact underlying gradient geometry of the objective function. Specifically, given a set of training samples $\{x_i, y_i\}_{i=1}^n$ where $x_i$ denotes the data and $y_i$ denotes the corresponding label, the training objective function and the full-batch gradient descent (GD) update at each step ($k = 0, 1, \ldots$) are written as follows.

$$\text{(Objective function): } \mathcal{L}_n(\theta) := \frac{1}{n} \sum_{i=1}^n \ell(h_\theta(x_i), y_i), \quad \text{(GD): } \theta_{k+1} = \theta_k - \eta \nabla \mathcal{L}_n(\theta_k), \quad (1)$$

where $h_\theta$ denotes the neural network model parameterized by $\theta$, $\nabla$ is the gradient operator with respect to the parameter $\theta$, $\eta$ is the learning rate, and $\ell$ is the loss function. We will consider classification tasks with the cross-entropy loss in this paper. In the training, we collect a set of gradients generated along the optimization trajectory of full-batch gradient descent, i.e., $\{\nabla \mathcal{L}_n(\theta_0), \nabla \mathcal{L}_n(\theta_1), \ldots, \nabla \mathcal{L}_n(\theta_k), \ldots\}$. These gradients determine the direction of model updates and help understand the local optimization geometry of the nonconvex objective function. To provide a quantitative understanding, we investigate the following pairwise gradient correlation of

any pair of gradients $(\nabla \mathcal{L}_n(\theta_j), \nabla \mathcal{L}_n(\theta_k))$ generated in the training:

$$\text{(Pairwise gradient correlation):} \quad \mu(j,k) := \frac{\langle \nabla \mathcal{L}_n(\theta_j), \nabla \mathcal{L}_n(\theta_k) \rangle}{\|\nabla \mathcal{L}_n(\theta_j)\| \cdot \|\nabla \mathcal{L}_n(\theta_k)\|} \in [-1, 1], \quad \forall j, k \quad (2)$$

where $\langle \cdot \rangle$ and $\| \cdot \|$ denote the inner product and the $\ell_2$-norm of vectors, respectively. At every epoch $k$, we fix a window size and compute the pairwise gradient correlations for the set of gradients $\{\nabla \mathcal{L}_n(\theta_k), \ldots, \nabla \mathcal{L}_n(\theta_{k+h-1})\}$. These gradient correlations form an $h \times h$ matrix, which illustrates the local geometry. We choose $h = 5$ for visualization purposes only. Moreover, at every epoch $k$, we also track the following mean gradient correlation $\overline{\mu}_k$ over adjacent epochs within the window.

$$\text{(Mean gradient correlation):} \quad \overline{\mu}_k := \frac{1}{h-1} \sum_{m=1}^{h-1} \mu(k + m - 1, k + m), \quad k = 1, 2, \ldots \quad (3)$$

We use $h = 5$ in eq. (3). The choice of $h$ is not essential as it only affects the smoothness of the mean gradient correlation curves.

## 3 MYSTERIOUS OPTIMIZATION GEOMETRY OF DEEP LEARNING

In this section, we train various modern deep networks and track the gradient correlation statistics introduced in Section 2. From these statistics, we consistently observe a mysterious and complex zigzag geometric pattern in practical deep learning optimization.

### 3.1 TRAINING CONVOLUTIONAL NETWORKS

**Simple CNN.** We first train a simple feed-forward CNN with three convolution blocks and one fully-connected block on the CIFAR-10 dataset (Krizhevsky, 2009) using full-batch gradient descent with learning rate $\eta = 0.1$ for 1500 epochs. We track the training loss, mean gradient correlation (defined in eq. (3)) and pairwise gradient correlation matrix throughout the training process. The results are shown in Figure 1. It can be seen that the training loss decreases to almost zero after 1000 epochs. On the other hand, the mean gradient correlation is highly positive for the first 10 epochs and drops to highly negative values after that. This shows that the gradients computed in adjacent epochs point toward the same direction only at the beginning of training and point toward opposite directions later on, implying a transition from a smooth geometry to a highly nonconvex geometry.

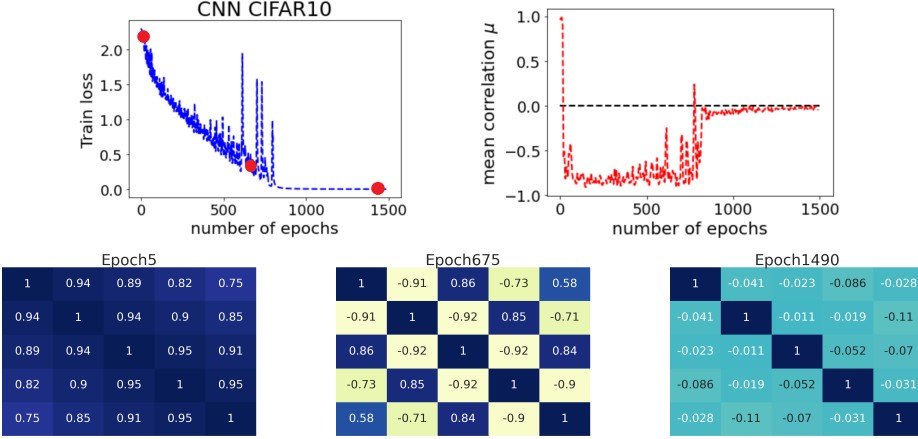

Figure 1: CNN training using gradient descent on CIFAR-10. Top row: training loss (left) and mean gradient correlation (right). Bottom row: pairwise gradient correlation matrices with window size $h = 5$ computed at epochs 5, 675, and 1490.

Moreover, the pairwise gradient correlation matrices reveal some interesting and mysterious geometric patterns in training. Due to space limitation, we present three representative pairwise gradient correlation matrices computed at epochs 5, 675, and 1490 (corresponds to the red dots in the training loss figure). We have the following observations from these matrices.

- **Epochs 1-10: smooth geometry.** In the first 10 epochs, the observed pairwise gradient correlation matrices are uniformly highly positive, as illustrated by the matrix computed at epoch 5 in Figure 1. This shows that in the initial phase of the training, the gradients computed within a small window of epochs are pointing in almost the same direction. Therefore, the optimization geometry of CNN is smooth in the initial training phase.

- **Epochs 10-700: zigzag geometry.** After 10 epochs, the algorithm's dynamic encounters a mysterious zigzag geometry, as illustrated by the pairwise gradient correlation matrix computed at epoch 675 in Figure 1. Specifically, the entries along the first off-diagonal line take negative values $\{-0.91, -0.92, -0.92, -0.9\}$ that are very close to $-1$. This implies that the gradients computed in the adjacent epochs (e.g., epochs 675 and 676) within this window are pointing in almost opposite directions. Also, the entries along the second off-diagonal line take positive values $\{0.86, 0.85, 0.84\}$ that are close to $1$. This implies that the gradients computed in the epochs with a lag order of two (e.g., epochs 675 and 677, epochs 676 and 678) within this window are pointing in similar directions. Furthermore, the entries along the third off-diagonal line become highly negative again, implying that the gradients computed in the epochs with a lag order of three (e.g., epochs 675 and 678) within this window are pointing toward opposite directions. These observations show that the optimization dynamics encounter a zigzag-type structure, and the optimization geometry of CNN is highly nonconvex and zigzag in the middle of the training.

- **Epochs 700-1500: random geometry.** When the training converges and achieves near zero loss after 1000 epochs, the pairwise gradient correlations are consistently close to zero, as illustrated by the pairwise gradient correlation matrix computed at epoch 1490 in Figure 1. This implies that the geometry of CNN is, to some extent, random around the global minimum.

**VGG-16.** We also explore the gradient correlation in training a VGG-16 network on the CIFAR-10 dataset using full-batch gradient descent with learning rate $\eta = 0.1$ for 1500 epochs. We observe similar optimization geometry to that of CNN. The results are shown in Figure 2. The mean gradient correlation is highly positive for most of the first 500 epochs. This is consistent with the pairwise gradient correlation matrix computed at epoch 5 shown in the figure, whose entries are uniformly highly positive. This shows that the gradients computed in the initial training phase are pointing in almost the same direction, therefore implying a smooth optimization geometry. However, in the later training phase after 500 epochs, the mean gradient correlation drops to highly negative values, indicating an occurrence of highly nonconvex geometry. In particular, we observe the zigzag geometric pattern in the pairwise gradient correlation matrix computed at epoch 915.

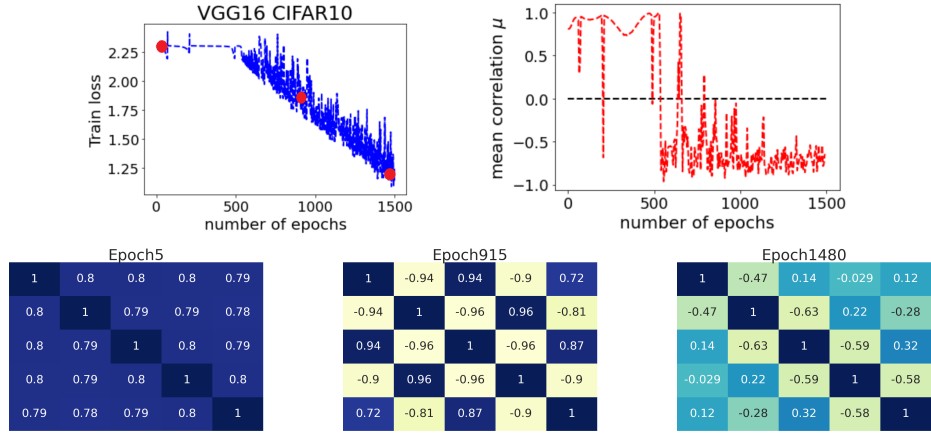

Figure 2: VGG-16 training using gradient descent on CIFAR-10

**Other results and conclusion.** In Appendix A, we include additional results on training the CNN and VGG-16 networks using other datasets such as MNIST (Deng, 2012) and SVHN (Netzer et al., 2011). From all these results, we observe a common phenomenon: the optimization geometry of CNN and VGG-16 has a sharp transition from smooth geometry to zigzag geometry in the training.

## 3.2 TRAINING RESIDUAL NETWORKS

We further train various residual networks and track the gradient correlation along the gradient descent trajectory. Interestingly, we also observe the zigzag geometry in training residual networks, but the overall transition of geometry is very different from that of convolutional networks.

**ResNet-18.** We train a standard ResNet-18 on the CIFAR-10 dataset using full-batch gradient descent with learning rate $\eta = 0.1$ for 500 epochs, and track the training loss and gradient correlation. The results are shown in Figure 3, from which one can see a very different transition of optimization geometry compared to that of convolutional networks. Specifically, one can see that the mean gradient correlation is highly negative in the first 120 epochs. This shows that the gradients computed in adjacent epochs are negatively correlated in the initial training phase, implying a highly nonconvex geometry. Moreover, from the pairwise correlation matrices computed at epochs 5 and 60, one can see that the local geometry gradually transfers to the zigzag geometry. After 150 epochs, the mean gradient correlation increases to highly positive values, implying a very smooth optimization geometry when approaching the minimizer. This is also supported by the pairwise correlation matrix computed at epoch 490, whose entries are uniformly close to 1. To summarize, the optimization geometry of ResNet-18 has a sharp transition from nonconvex zigzag geometry to smooth geometry. This is very different from the optimization geometry of convolutional networks, which starts with a relatively smooth geometry and transfers to zigzag geometry in the later training.

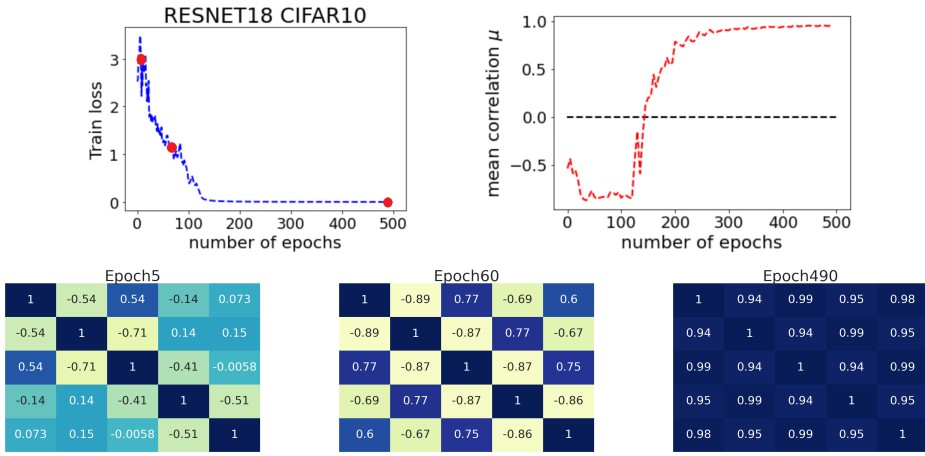

Figure 3: ResNet-18 training using gradient descent on CIFAR-10

**ResNet-34.** We further train a standard ResNet-34 on the CIFAR-10 dataset using full-batch gradient descent with learning rate $\eta = 0.1$ for 500 epochs. The results are shown in Figure 4, where one can observe a similar transition of geometry to that of ResNet-18.

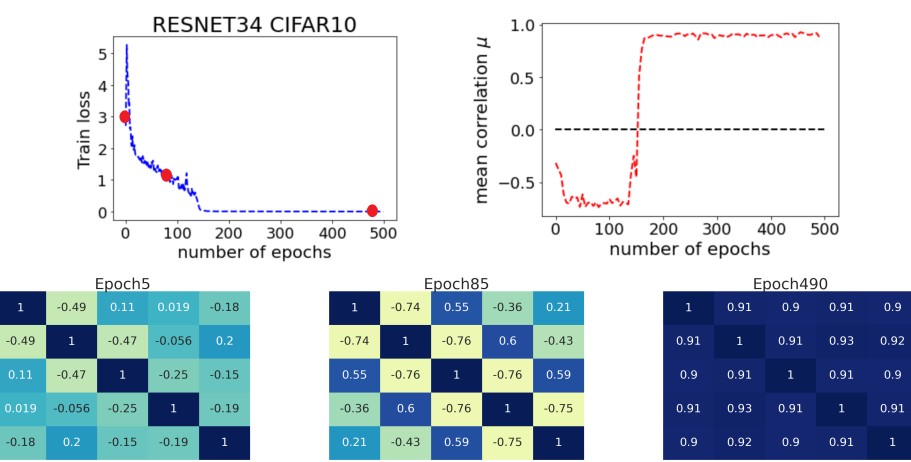

Figure 4: ResNet-34 training using gradient descent on CIFAR-10

**Other results and conclusion.** In Appendix B, we include additional results on training residual networks using MNIST and SVHN. From all these results, we observe a common phenomenon: the optimization geometry of residual networks has a sharp transition from nonconvex zigzag geometry to smooth geometry in training. On the other hand, in the existing literature, it has been visualized that the geometry of residual networks is smooth around the global minimum (Li et al., 2018b; Garipov et al., 2018). This is consistent with our observation that the mean gradient correlation is highly positive in the later training phase of residual networks. Our results further reveal that residual networks have complex zigzag geometry far away from the global minimum.

### 3.3 FRACTAL STRUCTURE OF ZIGZAG GEOMETRY

In the previous subsections, we observed the zigzag geometry in training various deep networks using a large learning rate. Here, we further zoom into the local geometry by training these networks with much smaller learning rates. We aim to understand if the zigzag geometry has any local structure. Figure 5 shows the training of VGG-16 on CIFAR-10 using full-batch gradient descent with a small learning rate $\eta = 0.01$. From the mean gradient correlation figure, it can be seen that the optimization geometry is smooth in the first 900 epochs, as also confirmed by the pairwise gradient correlation matrix computed at epoch 5. However, after 900 epochs, the mean gradient correlation instantly drops to highly negative values, and we start to observe the zigzag geometry as shown by the pairwise gradient correlation matrices computed at epochs 925 and 1490.

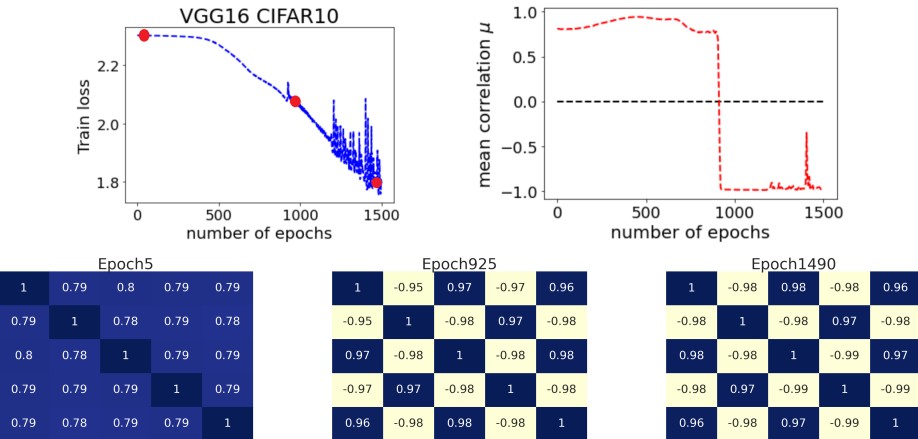

Figure 5: VGG-16 training using gradient descent with $\eta = 0.01$ on CIFAR-10

Figure 6 further shows the training of ResNet-18 on CIFAR-10 using full-batch gradient descent with a very small learning rate $\eta = 0.001$. It can be seen that the optimization geometry is very smooth for the initial 50 epochs. But it starts to transfer to zigzag geometry after 400 epochs, as can be seen from the pairwise gradient correlation matrices computed at epochs 490 and 990.

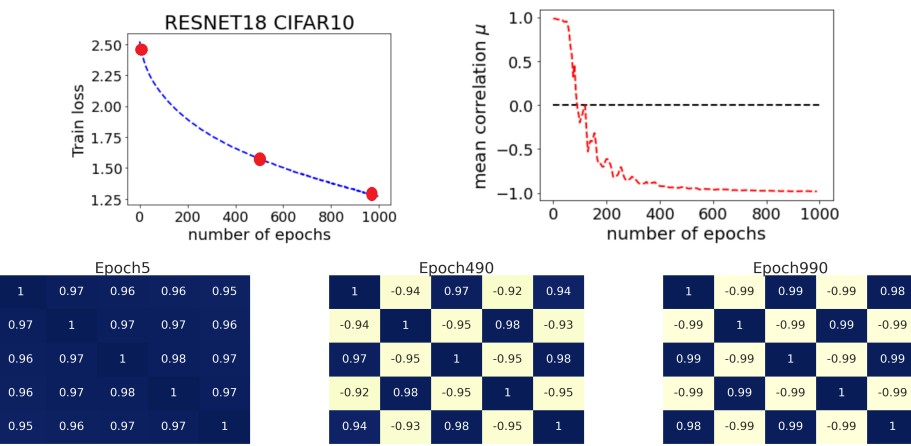

Figure 6: ResNet-18 training using gradient descent with lr=0.001 on CIFAR-10

**Other results and conclusion.** In Appendix C, we include additional results on training other models on CIFAR-10 using very small learning rates, and one can make very similar observations (namely, zigzag geometry still appears). From all these results, it seems that the zigzag geometry of deep networks has a complex fractal structure in that it appears within a wide range of geometrical scales. It indicates that deep networks can be highly non-smooth and non-Lipschitz in a local region of the parameter space. More interestingly, note that the variation of the entries of the zigzag pairwise correlation matrices in Figures 5 and 6 are much larger than those of the zigzag pairwise correlation matrices in Figures 2 and 3. This implies that the zigzag geometry tends to be more nonconvex when zoomed into a smaller geometrical scale.

## 4 DISCUSSIONS

### 4.1 DEEP LEARNING OPTIMIZATION THEORY

The observations made in Section 3 imply that many deep neural networks belong to a small class of functions with very special and nonconvex geometry, which is a mixture of smooth geometry and complex zigzag geometry. In particular, the gradients under the zigzag geometry constantly shift their directions at different geometrical scales. It implies a highly non-smooth local geometry: the local Lipschitz constants of the gradients are ill-conditioned and change rapidly.

On the other hand, this poses a challenge to the existing developments of deep learning optimization theory, which are often based on proving smooth and Lipschitz-type geometries of deep neural networks that guarantee convergence to a global minimum. For example, smoothness and gradient dominant geometry (also known as Polyak-Łojasiewicz geometry) constitute classic tools for establishing global convergence of gradient-type algorithms in nonconvex optimization (Karimi et al., 2016; Polyak, 1963). Specifically, the gradient dominant geometry requires the optimality gap of the objective function to be bounded by the corresponding gradient norm, i.e.,

$$\text{(Gradient dominant):} \quad \mathcal{L}_n(\theta) - \mathcal{L}_n(\theta^*) \leq C\|\nabla\mathcal{L}_n(\theta)\|^2, \quad \forall\theta \in \Theta,$$

where $C > 0$ is a universal constant, $\Theta$ is a local neighborhood of the minimizer $\theta^*$, and $\|\cdot\|$ denotes the $\ell_2$-norm. This nonconvex geometry has a close connection to other (non)convex geometries, including strong convexity (Nesterov, 2014), weak strong convexity (Necoara et al., 2019), and error bound (Luo & Tseng, 1993), all of which guarantee linear convergence of many gradient-based algorithms. In particular, many deep networks have been theoretically proved to have smooth and gradient dominant geometry, including over-parameterized residual networks (Zhang et al., 2019; Allen-Zhu et al., 2019b; Du et al., 2019a), some nonlinear networks (Zhou et al., 2016; Zou et al., 2020) and deep linear networks (Zhou & Liang, 2017; Frei & Gu, 2021). However, this is a coarse geometry model in practical deep learning, as it only involves the gradient norm information and does not capture the highly non-smooth zigzag geometry observed in our experiments. Therefore, we think the existing deep learning optimization theory cannot fully explain the success in practice, and advanced theories are needed to explain (i) why neural networks have such complex and non-smooth zigzag geometry and (ii) why gradient descent can overcome such non-smooth geometry and decrease the training loss in practice.

### 4.2 JUSTIFICATION OF LEARNING RATE SCHEDULING

Our previous experimental observations provide justifications for using learning rate scheduling in large-batch training of residual networks. Specifically, it is known that training residual networks with large batch size can easily diverge when a large learning rate is used in the initial training phase. To stabilize training, researchers proposed a warm-up learning rate scheduling scheme for large-batch training of residual networks (Goyal et al., 2017). The idea is to start with a very small learning rate and gradually increase to a large learning rate over several epochs. After that, the learning rate slowly decreases until the training is saturated (see the green curve in

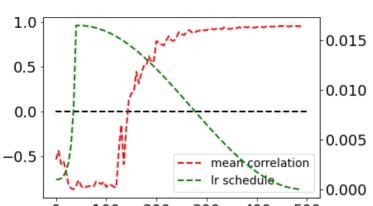

Figure 7: Red: mean gradient correlation of ResNet-18 training. Green: warm-up learning rate scheduling.

Figure 7). Our observation of mean gradient correlation in training residual networks justifies the effectiveness of this scheduling scheme. Specifically, as illustrated in Figure 7, the mean gradient

correlation in ResNet-18 training is highly negative in the initial training phase, which implies a highly nonconvex optimization geometry. Therefore, it is desirable to start with a small learning rate. After that, the gradient correlation gradually increases to highly positive values, and the geometry becomes very smooth. Therefore, a warm-up learning rate scheduling is appropriate. In Section 5, we propose to adapt the warm-up scheduling scheme to the mean gradient correlation to avoid parameter tuning in practice.

### 4.3 SIGN SHIFT OF GRADIENT ENTRIES

One interesting question is how many gradient entries shift their sign when encountering the zigzag optimization geometry. In Figure 8, we show the pairwise gradient correlation matrix when encountering the zigzag geometry in training convolutional networks. For each entry of pairwise gradient correlation, we calculate the percentage of the number of gradient entries with opposite signs. We can see that when encountering the zigzag geometry in CNN training, the gradient correlations of adjacent iterations stay around $-0.92 \sim -0.9$, and about $90\%$ of the gradient entries have opposite signs. This shows that such a complex geometry is across the majority of the parameter dimensions of convolutional networks. In Appendix D, we include additional results on sign shift of gradient entries in training residual networks, where one can make similar observations.

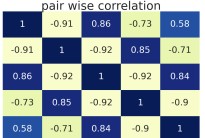 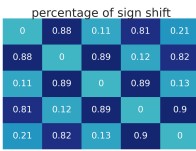 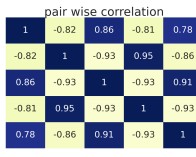 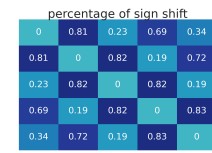

(a) Epoch 675 of CNN training on CIFAR-10      (b) Epoch 705 of VGG-16 training on SVHN

Figure 8: Gradient sign shift in training convolutional networks

## 5 APPLICATION: GEOMETRY-ADAPTED LEARNING RATE SCHEDULING

The statistics of gradient correlation sketch the global optimization geometry of deep networks. In this section, we further leverage this statistics to develop a *geometry-adapted* learning rate scheduling scheme for *large-batch training* of residual networks. We focus on large-batch training for two reasons: (i) it has become increasingly important and popular for large-scale training with parallel computation (Goyal et al., 2017; You et al., 2018; 2019), and (ii) the noisy stochastic gradient correlation can still reflect the underlying optimization geometry in the large-batch setting, as we show in the following subsection.

### 5.1 IMPACT OF BATCH SIZE ON GRADIENT CORRELATION

Figure 9 shows the mean gradient correlation in training a ResNet-18 with different large-batch sizes on CIFAR-10. In all the training, we linearly scale the learning rate based on the batch size. It can be seen that over a wide range of large-batch sizes, the mean gradient correlation curves do not change substantially. This implies that mean gradient correlation is a statistic that can reliably reflect the underlying optimization geometry in large-batch training. Please refer to Appendix E for more results on training other models, where one can make similar observations.

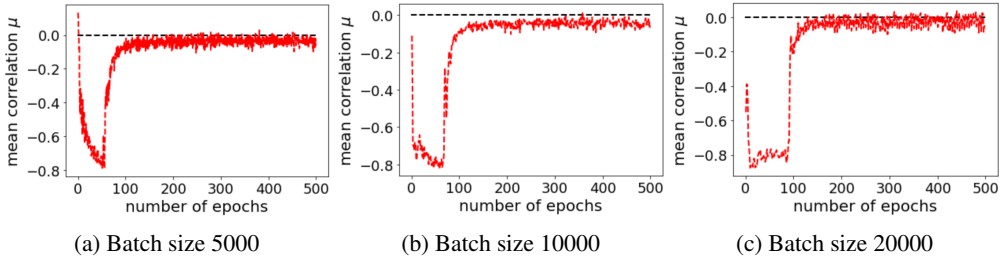

(a) Batch size 5000      (b) Batch size 10000      (c) Batch size 20000

Figure 9: Gradient correlation in ResNet-18 training on CIFAR-10 under different batch sizes

## 5.2 GEOMETRY-ADAPTED WARM-UP LEARNING RATE SCHEDULING

In training residual networks, the standard warm-up (WU) learning rate scheduling scheme consists of two phases, and the phase change iteration $t \in \mathbb{N}$ is heuristically chosen. In the warm-up phase (before $t$-th iteration), the scheme starts from a small learning rate $\eta_0$ and increases it exponentially as $\eta_k = \exp(\omega k)\eta_{k-1}$ for some hyper-parameter $\omega > 0$. In the decay phase (after $t$-th iteration), the scheme decays the learning rate via certain standard schemes (e.g., cosine annealing) until the end of the training. Here, we propose a *Geometry-Adapted Warm-Up* (GAWU) scheme that adaptively chooses the phase change point according to the mean gradient correlation statistics. Specifically, we keep tracking the mean gradient correlation in training and set the phase change point $t$ as the first time that the mean gradient correlation drops from positive to negative values, indicating a transition from smooth to non-smooth geometry. Furthermore, computing the mean gradient correlation is cheap in each iteration. It only requires storing an extra gradient calculated in the previous iteration and computing its correlation with the current gradient.

We compare the proposed GAWU scheme with the heuristic WU scheme in training a Resnet-18 on the CIFAR-10 dataset. We use SGD as the optimizer and set the batch size to be 5000. For the baseline WU scheme, we consider three heuristic change point settings $t = 5, 10, 12$, where 12 is the best change point that leads to the highest test accuracy that we found by performing a grid search over $t$. For both GAWU and WU, their warm-up phases use the same hyper-parameter $\omega = 0.1$, and their decay phases use the standard cosine annealing scheme with one period. We set the initial learning rate as $\eta_0 = 0.001$ for all the experiments.

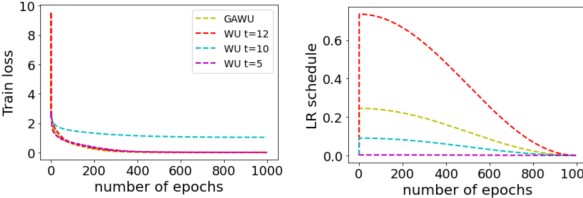

Figure 10: Comparison between GAWU and WU in training ResNet-18 on CIFAR-10.

| LR schedule | Test Acc (Loss) |
|---|---|
| WU ($t = 5$) | 61%(1.08) |
| WU ($t = 10$) | 74%(1.63) |
| WU ($t = 12$) | 78%(1.79) |
| GAWU | 76%(1.69) |

Table 1: Test accuracy of the final model trained by GAWU and WU.

Figure 10 shows the training loss and learning rate scheduling produced by GAWU and WU. It can be seen that the training loss obtained under the GAWU scheme has a comparable convergence speed to that of the training loss obtained under the WU scheme with fine-tuned parameter $t = 12$. In fact, in our experiment, the change point detected by GAWU is $t = 11$, which is very close to the best parameter $t = 12$ found empirically. This demonstrates the advantage of adapting the learning rate to the change of the underlying optimization geometry. Moreover, the test accuracy results presented in Table 1 further confirms the effectiveness of the GAWU scheme.

## 6 CONCLUSION

In this paper, we reveal that many popular deep networks have mysterious zigzag optimization geometry, which has a complex fractal structure over a wide range of geometrical scales. These observations imply that deep learning optimization cannot be fully characterized by the classic optimization theories, which crucially rely on elegant and smooth type geometries. As a future direction, we are motivated to develop a mathematical formalization of the observed zigzag geometry and develop advanced optimization theory to understand deep learning optimization in practice.

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
