# OpenReview forum: "On the Mysterious Optimization Geometry of Deep Neural Networks"
_ICLR.cc/2023/Conference — Submitted to ICLR 2023_

### Official Review · Reviewer_ynbp · 2022-10-24

**Confidence:** 3
**Correctness:** 4
**Technical Novelty And Significance:** 2
**Empirical Novelty And Significance:** 3
**Recommendation:** 5

**Clarity, Quality, Novelty And Reproducibility:**

**Quality**

The paper is very well-written and easy to follow. Unfortunately the plots in the figures appear a bit blurry on my monitor.

**Clarity**

Most parts of the paper are straight-forward to follow. It would have interesting to also include test accuracy/loss values to see whether the zigzag geometry also leads to an improvement there, or whether it is largely responsible for overfitting.

**Originality**

While gradient correlation is certainly not a new idea, the described zigzag geometry in non-toy settings is, to the best of my knowledge, novel.

**Strength And Weaknesses:**

**Strengths**

1. Understanding the loss landscapes of deep networks remains as an open and important problem. While theoretical works have made progress, it is often unclear how the underlying assumptions really reflect practical scenarios. Challenging these results through empirical experiments is very interesting and helpful for further understanding.

2. The setup is very simple and easy to understand. The authors study a very simple measure to assess the geometry of the landscape in the form of gradient correlations. The result of strong negative correlation over long periods of training is very surprising.

**Weaknesses**

1. While the gradient correlation measure is simple, it is not clear to me how well it reflects the simplicity of a loss landscape. The authors should present more evidence (empirical and theoretical) as to why negative correlation reflects difficulty in optimization, especially since training does make progress in terms of loss reduction comparable to the positively correlated stage. How do the gradients correlate if a convex problem is considered, i.e. linear regression? Are there theoretical results for this? How does negative correlation contradict gradient dominance and local strong convexity?

2. The shown figures for the correlation plots are extremely noisy (e.g. Fig. 1, Fig. 2, Fig 3.) which makes me assume that they are all based on a single run (I couldn’t find a statement indicating otherwise in the main text). More empirical evidence indicating that this is a robust phenomenon (i.e. averaging over multiple runs) would convince me more. While the authors argue that this geometry is robust w.r.t. learning rate, Fig. 3 and Fig. 6 seems to indicate differently. Here, a ResNet18 is trained with different learning rates, the large learning rate leading to an initial negative correlation while the smaller one to initial positive correlation. The large learning rate then transitions to positive correlation while the smaller one becomes negatively correlated. This seems to show that the metric is somewhat brittle and re-iterates my point for the need of multiple runs.

3. The previous point again makes me unsure regarding how well gradient correlation serves as a measure of roughness of a landscape. How can the nature of the landscape change for the same dataset and same model? For large learning rates, the landscape is highly non-convex first and then becomes easier (according to correlation) while for small learning rates, the landscape is very easy first and then becomes more difficult. The small learning rate experiments are also not very representative of the full landscape since we are very far from convergence even after 1000 epochs of training. Would the landscape remain easy until convergence or again transition after more training?

4. This work largely focuses on full batch gradient descent but how does this choice affect the obtained results, i.e. do we see any (negative) correlation if we were to use SGD instead and one would for instance consider the averaged gradient over one epoch? Fig. 9 shows some results for large batch sizes (smallest is 5000) but not for smaller, more practical sizes. Are you directly comparing gradients from mini-batches or averaging gradients first? The large number of epochs also seems untypical for the considered settings, is this due to full batch GD?

5. Finally I could also not find any attempt for an explanation of the zigzag phenomenon by the authors. Why would different learning rates affect the correlation in such a strong way? Why does a VGG exhibit such a different correlation trajectory, compared to a ResNet? How is it possible that the optimizer manages to reduce the training loss so well, even though effectively, gradients are simply just changing signs? Especially since the loss does not seem to reduce faster in the “well-behaved” possitively correlated setting over the same amount of time (see e.g. Fig. 2, Fig. 3, Fig. 5), some explanation would be very helpful.

**Summary Of The Paper:**

This work studies the optimization landscape of deep neural networks through tracking the  correlation of adjacent gradient steps of full batch gradient descent. Surprisingly, the authors find a strong zigzag behaviour, where adjacent gradients consistently have strong negative correlation values for a large period of training. Where this long stretch of zigzag behaviour occurs seems model dependent, the authors find that small CNNs and ResNets exhibit it close to the start of training, while VGGs show it towards the end of training. The zigzag behaviour indicates a difficult loss landscape (e.g. highly non-convex), which is in contrast to what prior work on gradient dominance and local strong convexity have identified. Using those insights, the authors develop a learning rate schedule for large batch training that takes the identified geometry into account. By tracking the mean correlation of gradients, the phase change point of the schedule is chosen to coincide with the time that the gradient correlation switches signs for the first time.

**Summary Of The Review:**

The observed zigzag phenomenon is interesting but as outlined above I am currently not convinced of the empirical evidence presented in this work. Multiple runs (at least for a key-setting, e.g. ResNet18 + CIFAR10) would help convince me of the robustness of this phenomenon. The authors should also motivate the metric better, showcase how in convex settings, no negative correlation emerges and discuss why different learning rates can completely flip the correlation trajectory. In general, more interpretation of the results is needed, why does a VGG have a different correlation trajectory than ResNets, why does the optimizer make similar progress in the same amount of time in terms of loss both in the difficult part of the landscape (negative correlation) and the easier part. In its current form I recommend rejection of this work. If the authors can convince me of the validity of their metric and provide better interpretation of their results, I am happy to change my score.

---

### Official Review · Reviewer_mFYm · 2022-10-24

**Confidence:** 4
**Clarity, Quality, Novelty And Reproducibility:** The paper is clear
**Correctness:** 2
**Technical Novelty And Significance:** 2
**Empirical Novelty And Significance:** 2
**Recommendation:** 3

**Strength And Weaknesses:**

Strengths

- The optimization geometry of deep learning is not well understood, and this paper provides a different perspective.

- Correlation-dependent learning rate scheduling seems novel.


Weaknesses

- This work does not have a clear hypothesis to test.
It seems that the authors implicitly hypothesized that gradient directions should be consistent in subsequent time steps.
Instead, they found that gradients may be anti-correlated and thought that this is surprising.
However, there does not seem to be any fundamental reason to believe that gradients should be consistent.

- In fact, the observation of anti-correlated gradients can be reproduced even in simple toy models.
Take, for example, a quadratic loss with Hessian h, anti-correlated gradients would be observed when the learning rate l is 1/h < l < 2/h.
In high-dimensional scenarios, different directions in parameter space have different (eigen-)values of the Hessian, and for some (perhaps a large fraction) of them anti-correlation should be expected.
Most importantly the Hessian changes during training, and in some circumstances it keeps increasing, so much that anti-correlations should be expected even if the learning rate is small.
This phenomenon is studied in detail in this paper: https://arxiv.org/abs/2103.00065, which may explain several of the author's observations.

- The authors contrast anti-correlated gradients with "smooth" behavior.
In fact, they even state that "the mean gradient correlation drops from positive to negative values, indicating a transition
from smooth to non-smooth geometry."
However, gradient anti-correlation does not imply non-smoothness of the loss function, as the simple quadratic example suggests.
(I assume the usual definition of smoothness, for twice differentiable functions, that the Hessian is bounded.)

- It is true that, if the anti-correlation of gradients would persist in the limit of zero learning rate, then it would imply non-smoothness of the loss.
However, this limit is not shown in the paper, only a couple of values of the learning rate are considered, which is very far from proving any "fractal structure".
Even if the Hessian increases during training, it is expected to remain bounded (see e.g. http://arxiv.org/abs/2204.11326).

- I found that the most interesting part of the paper is about the correlation-dependent learning rate scheduling, section 5.
However, very limited results are shown and it is very hard to draw any conclusions.



**Summary Of The Paper:**

The goal of this paper is "investigating the optimization geometry of deep networks".
In particular, correlations between gradients in adjacent time steps is measured in computer vision tasks (mostly convnets on CIFAR).
It is observed that correlations during training have periods of positive values, implying that the gradient points at a similar direction in consecutive time steps, and periods of negative values, implying that the gradient points at a nearly opposite direction in consecutive time steps.
A learning rate scheduling that depends on gradient correlation is proposed and preliminary results are shown.



**Summary Of The Review:**

Most of the results of this paper are not surprising and can be interpreted in the light of simple models (e.g. quadratic loss) or more recent findings (e.g. https://arxiv.org/abs/2103.00065).
The section on correlation-dependent learning rate scheduling is interesting and novel, but shown results are very limited and unconvincing.

---

### Official Review · Reviewer_MfSA · 2022-10-25

**Confidence:** 3
**Correctness:** 3
**Technical Novelty And Significance:** 3
**Empirical Novelty And Significance:** 3
**Recommendation:** 6

**Clarity, Quality, Novelty And Reproducibility:**

The paper is written clearly and contributions are novel, so not much to point in this regard.

**Strength And Weaknesses:**

In terms of strenghts:
- In CIFAR-10, their analysis is quite throughout, exploring variations in architecture and optimization regime.
- Their proposed geometry-aware learning rate scheduler shows that their methodology and findings have potential to make optimization of neural networks more ammenable

However my main concern is if these findings regarding the geometry of the loss landscape are “universal” or are instead a peculiarity of the setting used, and if they apply to real use cases of the models.
    - My main worry is that the authors only train model CIFAR-10. I understand that CIFAR-10 makes it easier to study the full-batch case, but, given their findings (5.1), I think studying how the gradient correlations evolve in other, more realistic, image dataset (say ImageNet) under the stochastic gradient descent case would be important to understand if this happens in realistic cases. I think this make the paper much more impactful.
    - The authors only study CNNs and CNNs with ResNet connections. In contrast, most of the current advances (to my knowledge) are done with transformer-like models. Given the stark differences in architectures, I am really curious to know if their methodology based on gradient correlation can be applied to transformer-like architectures and if they loss geometry evolves in similar manner.


**Summary Of The Paper:**

This paper studies how geometry of the loss landscape changes throughout training by studying the how gradient vectors of consecutive training steps correlate with each other: high correlations across timestep windows imply a smooth geometry while alternating gradient directions imply a non-convex, “zigzag” geometry.

The authors study how these correlations evolve by training different architectures (CNNs and ResNets) on the CIFAR-10 image classification dataset. Under the full-batch setting, they find that the geometry of the loss landscape has distinct phases throughout training, and that depend on the architecture used:
- For CNN architectures it has (1) an initial smooth phase, where consecutive correlate with each other, followed by a (2) zig-zag phase where gradients alternate directions and, when saturating the training loss, ending in a (3) random geometry phase, where gradients don’t correlate with each other
- For ResNet architectures, the loss landscape instead starts in (1) zig-zag geometry and evolving throughout training to a (2) smooth geometry, even after the learning plateaus.

The authors also claim to observe an interesting phenomena of **fractal** geometry, where the same evolution of geometry are observed a smaller geometric scale when training with smaller learning rates.

They also discuss how their observations might justify the success of learning rate schedulers, such as warmup, in networks with residual connections: The lower learning rates are help-up in mitigate the non-convex, zig-zag geometry these networks have early in their training. The authors then propose a learning rate schedules that explicitly leverages their notion of geometry to automatically determine how many steps to do warm-up with, based on the phase changes in geometry.

**Summary Of The Review:**

This paper provides a novel analysis of the optimization geometry of neural networks using gradient correlation across consecutive steps. While the analysis is throughout and novel, and brings new insights about optimization landscape of neural networks, they only study optimization on the CIFAR-10 dataset, which makes me worry about the generality of their results.

---

### Author Response · Authors · 2022-11-13
**Summary response to the provided review**

We thank all the reviewers for reviewing our manuscript and providing valuable feedback. Below we provide a summary response to the major concerns of the review comments.

**Q:** Concern about the generality of the results (used only CIFAR-10 in full batch setting). Suggest to run multiple repetitions of the experiments.

**A:** Regarding the first concern, we do have similar and coherent results obtained on SVHN and MNIST datasets alongside other models such as VGG-16 and Alexnet (see Appendices A and B). We also presented additional results that use different batch sizes in the Appendix E. In the future revision, we will add more repetitions of the experiments, thanks for the suggestion.

**Q:** In (Cohen et al. 2021), the Hessian changes during training and in some circumstances it keeps increasing so much that anti-correlations should be expected.

**A:** Thanks for pointing out this interesting reference. We think the sharpness (i.e., the maximum eigenvalue of the Hessian) along does not fully characterize the gradient behavior during the training, especially in the highly nonconvex deep learning case. In their Figure 2, they did provide a synthetic convex example for which gradient descent oscillates. But this example only considers a convex quadratic function with synthetic data and high sharpness, and their focus was not gradient correlation but sharpness. On the other hand, as suggested by one of the reviewers, it will be interesting to explore if similar negative correlation can be observed in training convex models (e.g., one-layer neural network) on real datasets. We will cite this paper in the revision and further discuss their connections and differences.

**Q:** More evidence as to why negative correlation reflects difficulty in optimization? How does negative correlation contradict gradient dominance and local strong convexity? Should also showcase how in convex settings, no negative correlation emerges.

**A:** Great question. For example, consider the strongly convex case that satisfies $\langle \nabla f(x) - \nabla f(y), x-y \rangle \ge \mu \|x-y\|^2$ for all $x,y$. Suppose negative gradient correlation holds. We can update $y$ to $\tilde{y}$ via one gradient descent step with a very small learning rate so that $x-\tilde{y}$ is almost unchanged. However, $\nabla f(\tilde{y})$ can be opposite to $\nabla f(y)$ due to negative correlation, and therefore the strong convex inequality may be violated between $x$ and $\tilde{y}$. We will add some convex experiments to show that such a phenomenon cannot be observed in the convex case. Thanks for the suggestion.

---

### Decision · Program_Chairs · 2023-01-20

**Decision:**

Reject

**Justification For Why Not Higher Score:**

A number of reviewers concerned about how the negatively correlated gradient in adjacent time step can reflect the non-smooth landscape. There are no experiments on large-scale datasets (like ImageNet).

**Justification For Why Not Lower Score:**

N/A

**Metareview: Summary, Strengths And Weaknesses:**

This paper studies the optimization geometry, more specifically, the gradient behaviors of VGG and ResNet. The conclusion in this paper has been studied in existing works and a number of reviewers concerned about how the negatively correlated gradient in adjacent time step can reflect the non-smooth landscape. There are no experiments on large-scale datasets (like ImageNet). The rebuttal did not address the reviewers' concerns. For these reasons, I recommend reject.